



# Studies on the Propagation Dynamics and Source Mechanism of Quasi-Monochromatic Gravity Waves Observed over São Martinho da Serra (29°S, 53°W), Brazil

Cristiano M. Wrasse[1], Prosper. K. Nyassor[1], Ligia A. da Silva[1,2], Cosme. O. A. B. Figueiredo[1], José V. Bageston[3], Kleber P. Naccarato[4], Diego Barros[1], Hisao Takahashi[1], and Delano Gobbi[1]

[1]National Institute for Space Research, Space Weather Division, São José dos Campos, SP, Brazil
[2]State Key Laboratory of Space Weather, National Space Science Center, Chinese Academy of Sciences, São José Dos Campos, SP, Brazil
[3]Southern Space Coordination, Santa Maria, National Institute for Space Research, RS, Brazil
[4]Impacts, Adaptation and Vulnerabilities Division, INPE, São José dos Campos, SP, Brazil

**Correspondence:** Cristiano Max Wrasse (cristiano.wrasse@inpe.br)

**Abstract.** Two hundred and nine (209) events of quasi-monochromatic atmospheric gravity waves (QMGWs) were acquired over five (5) years of Gravity Waves (GWs) observation in Southern Brazil. The observations were made using OH all-sky imagers hosted by the Southern Space Observatory (SSO) coordinated by the National Institute for Space Research at São Martinho da Serra (RS) (29.44°S; 53.82°W). A two (2) dimensional Fast Fourier Transform-based spectral analysis shows that the QMGWs have horizontal wavelengths of 10 - 55 km, periods of 5 - 74 minutes, and phase speeds up to 100 m/s. The waves exhibited clear seasonal dependence on the propagation direction with anisotropic behavior, propagating mainly toward the southeast during the summer and autumn seasons and mainly toward the northwest during the winter. On the other hand, the propagation directions in the spring season exhibited a wide range from northwest to south. A complimentary backward ray tracing result revealed that the significant factors contributing to the propagation direction of the QMGWs are their source locations and the dynamics of the background winds per season. Three case studies in winter were selected to investigate further the propagation dynamics of the waves and determine their possible source location. We found that the jet-stream associated with cold front and their interaction generated these three GW events.

## 1 Introduction

Atmospheric gravity waves (GWs) have earned research interest for decades due to their significant role in energy and momentum transportation throughout the atmosphere. GWs can be generated by various sources: orography, jets, and deep convection, such as thunderstorms, in the lower atmosphere. GWs with various spectra simultaneously exist in the atmosphere with several propagation characteristics (Wei and Zhang, 2014; Zhang et al., 2015). GWs with almost the same spectrum in relation to their spatial or temporal characteristics are known as quasi-monochromatic (QM) GWs. Studies on QMGWs can be used to reveal the relationship between the scale of the observed wave, the propagation characteristics, and the source-source mechanism.





QMGWs are frequently observed with airglow imagers, lidars, and radars. The QMGWs observed by imagers typically have a short horizontal wavelength ($\lambda_H$) and high frequency (Hecht et al., 2001; Walterscheid et al., 1999), while those observed by radars and lidars typically have long $\lambda_H$ and low frequency (Gavrilov et al., 1996). All sky imagers (ASIs) have been widely used in GW observations since Peterson and Adams (1983) first observed wave perturbations in the OH airglow.

The wave characteristics of GWs are essential information needed to determine/anticipate the possible source or source

mechanism of the observed wave. In most cases, spectral analysis techniques based on two-dimensional (2D) fast Fourier transform are used to estimate GWs parameters for observations made using an imaging technique. Among others, Vadas et al. (2009, 2012), Paulino et al. (2012), and Nyassor et al. (2021) used spectral analysis in two dimensions to determine parameters of observed GWs using ASI in the OH emission layers. They used the wave parameters as inputs in a ray tracing model to determine the possible wave source in their work. Nyassor et al. (2021) further investigated the source mechanism based on the

source location determined by ray tracing. Lai et al. (2020) used a discrete wavelet transform (DWT) based algorithm followed by a denoising and an adaptive scan band-pass filter procedures to estimate the propagating characteristics of the GWs of different scales observed by a network of ASI.

This work investigates the propagation, source location, and source mechanism of QMGWs observed using ASI. Spectral analysis was used to determine the gravity wave parameters used as input in the ray tracing model to investigate the propagation

of the QMGWs in the atmosphere and to determine the source location. Also, three case studies were conducted to investigate the wave sources. We found that the three case studies showed peculiar characteristics in their propagation direction with time. Finally, it was also found that the interaction between the cold front and jet streams excited these three gravity waves.

## 2   Observation

### 2.1   OH All-Aky Imager

Gravity wave observations were taken at the Southern Space Observatory (SSO), located in São Martinho da Serra (SMS) (29.44°S; 53.85°W), Rio Grande do Sul, Brazil, using an all-sky imager. The observatory belongs to the National Institute for Space Research (INPE) under the Southern Space Coordination (COESU/INPE).

The all-sky imager is equipped with a Charge Coupled Device (CCD) camera (model STL-1001E by SBIG), which has a resolution of 1024 × 1024 pixels, each pixel measuring 24.6 × 24.6 mm. It also comprises a fisheye lens, a telecentric lens

system, and an objective lens. The instrument uses a single filter for OH observations (715 - 930 nm, with a notch at 865.5 nm) which originates from a ~7 to 8 km thick layer located at ~87 km altitude. Details on the observation mode (including the temporal resolution and integration time) and production of the final image can be seen elsewhere in Bageston et al. (2009) and Nyassor et al. (2021, 2022).





## 2.2 Geostationary Operational Environmental Satellite (GOES)

The Geostationary Operational Environmental Satellite (GOES) - R Series is used to observe and study tropospheric convection. The infrared channels of the Advanced Baseline Imager (ABI) are used in this work. This channel has a spatial resolution ranging from 0.5 to 2 km and a temporal resolution of 10 min that allows the observation of critical weather and climate products for Full Disk and mesoscale. The cloud top brightness temperature (CTBT) product is derived from the 11, 12, and 13.3 $\mu$m infrared observations. These channels are used in the observation of cloud-top and cold-front activities.

## 3 Methodology and Data Analysis

Original all-sky airglow images with curvature effects contain both wave events and stars, over the field of view of the image during clear sky conditions. Therefore, it is important to preprocess the images before the determination of the spectral characteristics of the observed waves. The image preprocessing and spectral analysis procedure follow the procedure of Garcia et al. (1997) and Wrasse et al. (2007).

The preprocessing begins with the alignment of the original airglow image to the geographical north (thus, to a standard coordinate), after which the stars are removed. The intensity of airglow observed by a ground-based imager is not uniform due to varying zenith angles, even for a spatially uniform airglow emission. The observed intensity is proportional to the length of the line of sight (LOS) in the airglow emission layer, known as the van Rhijn effect. Also, the amount of absorption by the atmosphere is proportional to the length of the LOS from the emission layer to the observation point. This absorption by the atmosphere, known as atmospheric extinction, weakens the observed airglow intensity. The Van Rhijn effect and atmospheric extinction were corrected by applying the method of Kubota et al. (2001). Afterward, the images are unwarped and mapped onto the geographical coordinates.

The gravity wave parameters: horizontal wavelength ($\lambda_H$), phase speed ($c_H$), observed period ($\tau$), and propagation direction ($\phi$) are then determined using the 2D Discrete Fast-Fourier transform (2D-DFT) based spectral analysis (Garcia et al., 1997; Wrasse et al., 2007). Before the application of the 2D-DFT, regions of interest (ROI) with visible waves (clear dark and bright bands) were then selected. Afterward, a time series of 10 images was constructed with the selected ROI, and the 2D-DFT was applied to the ROI in the selected image time series. From the cross-spectrum of the 2D-DFT, the amplitude and phase of the wave are estimated and used to calculate the $\lambda_H$, $c_H$, $\tau$, and $\phi$. Details on the spectral analysis can be found in Wrasse et al. (2007), Bageston et al. (2011), Giongo et al. (2020), and Nyassor et al. (2021). The image preprocessing, spectral analysis and estimation of the gravity wave parameters are summarized in a flowchart in Figure 1.





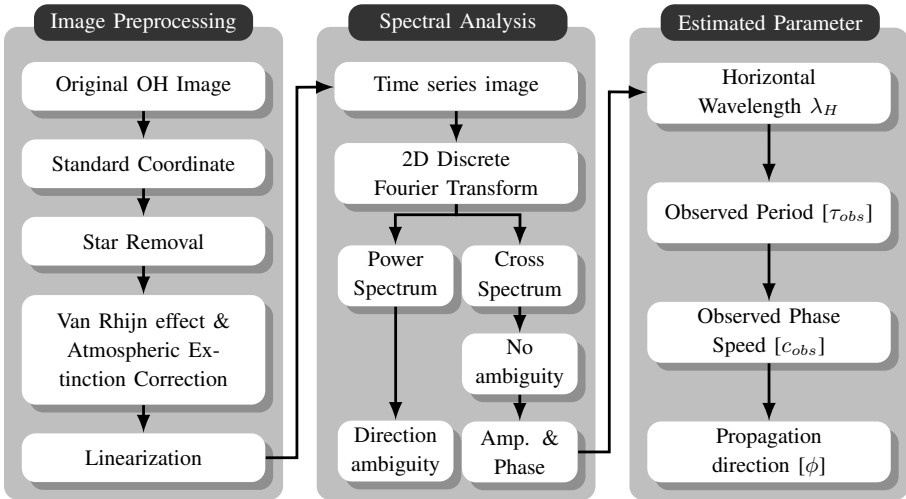

**Figure 1.** Flowchart showing the procedures of airglow image processing and wave parameter estimation. The three stages describe image preprocessing and processing, spectral analysis, and wave parameters estimation procedure.

## 3.1 Ray Tracing Model

A ray tracing model was used to investigate the QMGWs' propagation conditions and their source locations. In this work, the ray tracing model follows the approach of Vadas (2007), Paulino et al. (2012), and Nyassor et al. (2021), with the underlying formalism from Lighthill (1978). However, in this version of the ray tracing model, kinematic viscosity and thermal diffusivity (Vadas, 2007) are incorporated in the group velocities, and the dispersion relations are similar to the work of Vadas and Fritts (2005). The longitude, latitude, and altitude (at 87 km) of the first visible crest/trough and the observation time of the wave are assumed as the initial positions and times of the wave, and the wave characteristics are then used as the input parameters for the model.

The next step, thus, in longitude, latitude, altitude, and time of the iteration, which formed six ordinary differential equations, was solved using the fourth-order Runge-Kutta (Press et al., 2007). An initial altitude step size of 0.2 km was set, and the subsequent step sizes were determined from $z = c_{g_z} t$, with the boundary conditions of Paulino et al. (2012) imposed. The next step iteration is conducted if the following criteria are satisfied:

1. The group velocity of the GWs must be less than or equal to 0.9 times the speed of sound ($c_g \leq 0.9Cs$).

2. To evaluate the effect of the background wind on the wave propagation, the real component of the intrinsic frequency must be greater than zero ($\omega_{Ir} > 0$).

3. The momentum flux along the wave trajectory is evaluated in relation to molecular viscosity and thermal diffusivity since they become important dissipative processes with increasing altitude. GWs tend to dissipate when they attain maximum momentum flux; therefore, for a propagating GW, $R_m > 10^{-15} R_0$. Here $R_m$ is the momentum flux at each altitude, and $R_0$ is the momentum flux at the reference altitude. The factor $10^{-15}$ was arbitrarily chosen.



4. To ensure slowly varying viscosity in time and altitude, the module of the vertical wavelength must be less than the viscosity scale $\left[\mid \lambda_z \mid < 2\pi / \frac{dv/dz}{\nu}\right]$ in that $\nu = \mu\rho$ is kinematic viscosity, where $\rho$ is molecular viscosity and $\rho$ is density (Nyassor et al., 2022). The value of $\mu = 3.34 \times 10^{-4} T^{0.71}$, increases with altitude, where $T$ is temperature Vadas (2007).

The items (3) and (4) are important for studying GW propagating into the thermosphere. If there is a violation of the above-defined criteria, the iteration will be interrupted, and then all the calculations end and are saved automatically. The stopping conditions are discussed in Vadas (2007) and Paulino et al. (2012).

Atmospheric background wind and temperature used in the ray tracing were obtained from the Modern-Era Retrospective and analysis for Research and Application-version 2 (MERRA-2) data (Gelaro et al., 2017), the Horizontal Wind Model 2014 version (HWM14) (Drob et al., 2015), and the Mass-Spectrometer-Incoherent-Scatter (NRLMSISE-00) empirical atmospheric model (Picone et al., 2002). Due to the limited altitude range of MERRA-2 wind and temperature data, which is up 75 km, we concatenated the MERRA-2 wind data with HWM14 at an interpolated step at each 1 km. Similarly, the temperature data of MERRA-2 and NRLMSISE-00 are also concatenated. This procedure is done to attain an altitude range from near the surface of the ground up to 100 km. Since two different datasets with different resolutions are being concatenated, there may exist discontinuities at the concatenation altitude. The discontinuities are minimized by using the approach of Nyassor et al. (2022). As a result of the temporal resolution of MERRA-2, which is 3 h, an interpolation was performed for each time step of the ray tracing iteration. The propagation of the wave through the atmosphere leading to the determination of the source location of the wave is investigated using ray tracing in a backward mode.

## 4 Results

### 4.1 Observed QMGWs

Observations of QMGWs began in April 2017 and ended in April 2022 in São Martinho da Serra. A total of 64 QMGW cases were acquired from 1512 nights of airglow observations. The monthly distribution of observed QMGWs is presented in Figure 2. Each bar shows the accumulated QMGW cases observed each month, with the individual colors in gray showing the number of events observed each year. The color bar defines the year of observation. It can be seen from Figure 2 that the highest number of QMGW cases was observed in August, followed by July. The following section will discuss details on the distribution of the QMGW cases in Figure fig:figure02.





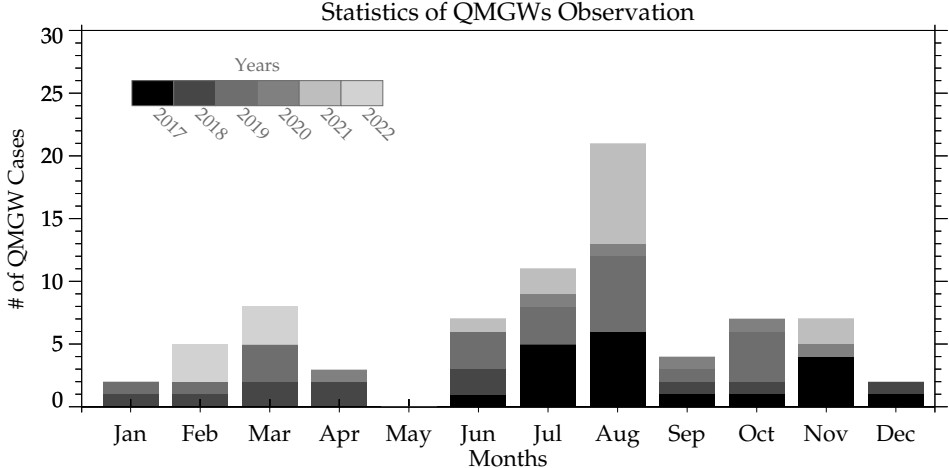

**Figure 2.** The observed quasi-monochromatic gravity wave (QMGW) cases were distributed between 2017 and 2021 at São Martinho da Serra.

## 4.2 Statistical Distribution of QMGW Events and Parameters

The five years of observed OH airglow images were subjected to spectral analysis to estimate the QMGW characteristics. Specific criteria were imposed to select the QMGW events used in this work. After the spectral analysis, the confidence level (CL) of the estimated wave spectrum is estimated. The spectrum having peak power spectral density with CL $\geq 95\%$ is accepted (Hu et al., 2002). With the CL $\geq 95\%$ condition being satisfied, the wave is then subjected to the following conditions:

  i. the $\lambda_H$ must be greater than or equal to 10 km ($\lambda_H \geq 10$ km);

  ii. the variation of $\phi$ within an hour must be less than 25° ($\Delta\phi \leq \pm 25°$);

  iii. the GW propagation period must be between 5 and 80 min (5 min $\geq \tau \leq 80$ min), and

  iv. finally, the GW phase speed must be less than or equal to 150 m/s ($c_H \leq 150$ m/s).

Note that the $c_H > 150$ m/s was considered the upper limit to avoid interference with the acoustic wave spectrum. Vadas and Azeem (2021) mentioned that GWs with $c_H \sim 250$ m/s cannot propagate below 100 km. However, we chose this value since it will take a wave with 150 m/s, approximately $\sim 12$ minutes, to travel 100 km. If all these conditions are satisfied, the wave is selected. On the contrary, the wave will be removed even if CL $\geq 95\%$ and one of the other conditions are violated.

Figure 3 presents the characteristics of the selected QMGW parameters obtained from the spectral analysis. Panel (a) shows the QMGWs horizontal wavelength distribution over the five years of observation. Panel (b) is the distribution of the propagation period, whereas panel (c) is the bar plot of the phase speed distribution. Panel (d) shows the distribution of the propagation direction of the QMGWs. The solid black lines in panels (a), (b), and (c) show the Gaussian fit to give an idea about the general distribution of the wave parameters.





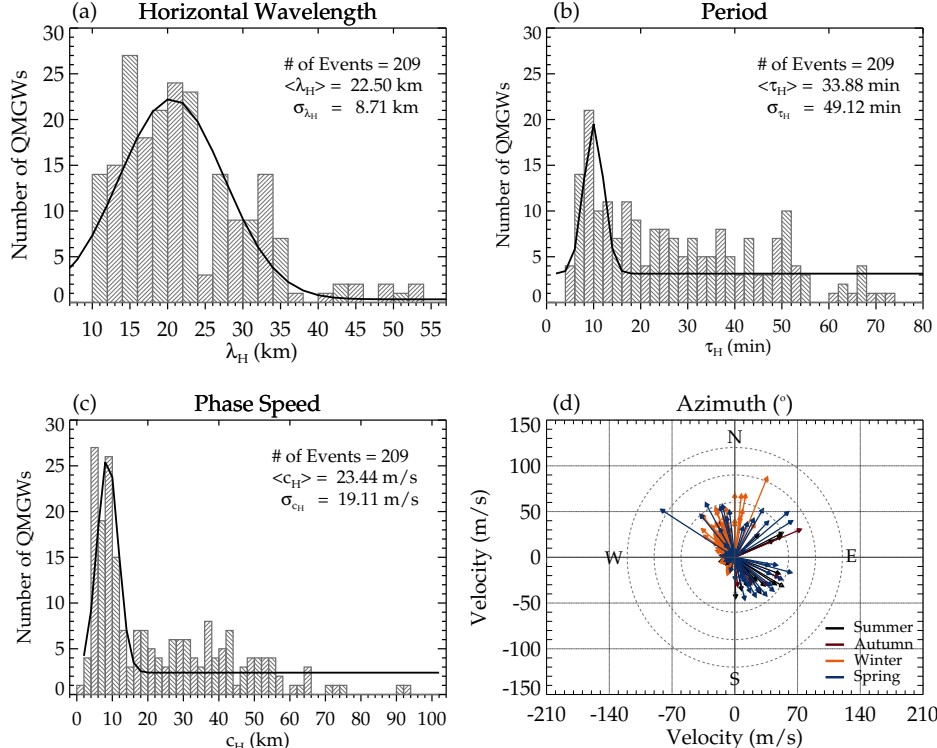

**Figure 3.** Quasi-Monochromatic Gravity Waves (QMGWs) characteristics over five years of observations at São Martinho da Serra. Panels (a), (b), (c), and (d) present the occurrence rate of the horizontal wavelength ($\lambda_H$), period ($\tau_H$), phase speed ($c_H$), and propagation direction ($\phi$), respectively.

For the $\lambda_H$ distribution in panel (a), an average wavelength of 22.50 km was observed with a peak value of ∼15 km. The Gaussian fit indicated by the solid black line shows that the broad and dominant distribution lies between 10 and 35 km.

However, the normal distributions of the $\lambda_H$ (panel (b)) and $c_H$ (panel (c)) are narrow with a dominant peak period, and phase speed skewed toward ∼10 minutes and ∼9 m/s, respectively. The propagation direction of the QMGWs is presented in panel (d) of Figure 3. The direction of wave propagation is significantly anisotropic, mainly between northwest to northeast, during the summer and southeast directions during the winter.

## 4.3 Ray Tracing Results

Ray tracing model is used to study the propagation of the QMGWs and determine their possible source locations. Two wind models were considered when running the ray tracing model: zero wind and model wind modes. The model wind mode consists of concatenated MERRA-2 and HWM14 wind. However, in this work, only the model wind result of the ray tracing is presented. The ray tracing results for the 209 QMGWs are presented in Figure 4. The ray paths of the QMGWs in a model





wind atmosphere are shown in blue lines, and their respective stopping positions are in red squares. The open triangle shows

the observation site location.

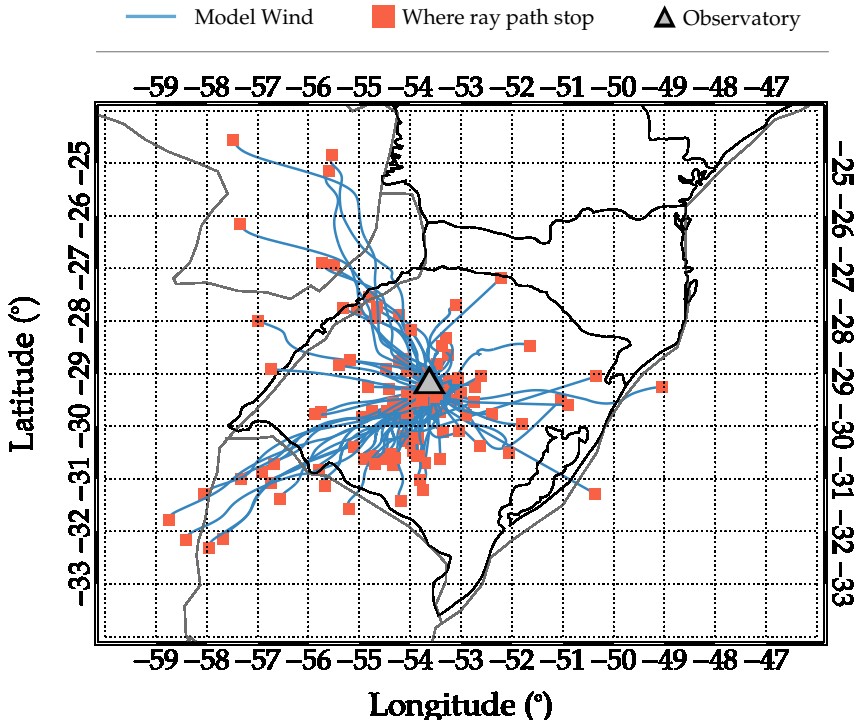

**Figure 4.** Backward ray paths and stopping positions of the observed Quasi Monochromatic Gravity Waves (QMGWs) over São Martinho da Serra.

The propagation time of the waves from their source to the observation altitude in the mesosphere is presented in Figure 5(a), while the duration of propagation of the waves in the OH images is presented in Figure 5(b). It can be seen that the majority of the wave propagated less than one hour from the source position in the lower atmosphere to the OH emission layer. Similarly, most waves observed in the OH airglow images were visible and propagated for 2 - 3 hours.



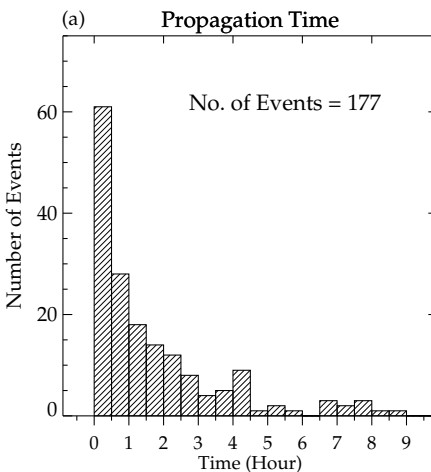
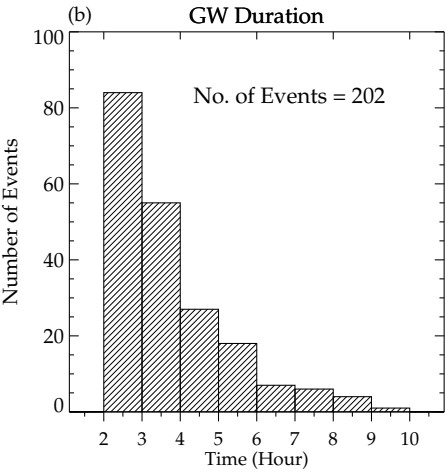

**Figure 5.** Propagation and visibility times of the Quasi Monochromatic Gravity Waves (QMGWs). (a) The propagation time of the wave from the source position to the OH emission layer. (b) The duration of propagation of visible QMGWs in the OH images.

### 4.4 Wave Sources

Most mesospheric GWs have their primary sources in the lower atmosphere. Various generation mechanisms, such as the mechanical oscillator effect, obstacle effect, and latent heat of deep convection and orographic, are known to be prominent source mechanisms (Fritts and Alexander, 2003). From the ray tracing result presented in Figure 5, it was observed that 12.4% of the ray path stopped above 60 km, implying these waves are generated in situ. However, the source mechanisms of these waves will not be discussed in this current work. On the other hand, the ray path of the remaining 87.6% stopped in the troposphere. It indicates that this percentage of the wave is most probably generated in the troposphere. Figure 6 presents the distribution of the minimum cloud top brightness temperature near the stopping positions of the ray paths in the troposphere.



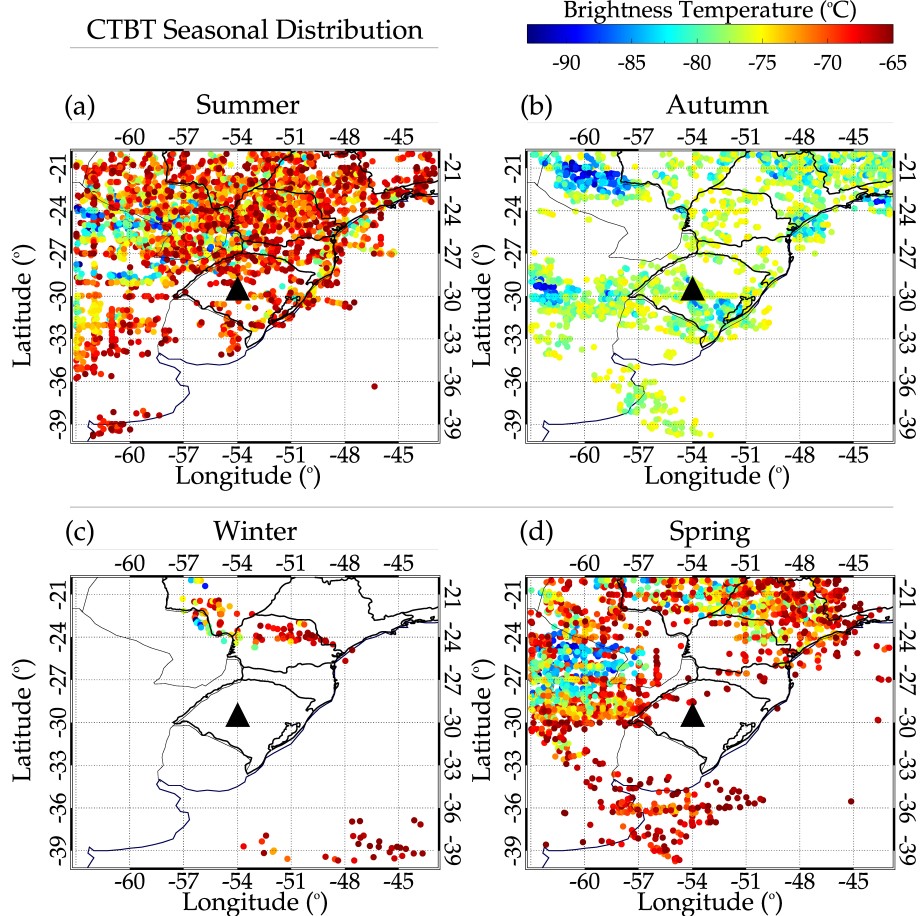

**Figure 6.** Seasonal distribution of the cloud top brightness temperature (CTBT) close to the stopping locations of the ray paths in the tropopause.

Panels (a), (b), (c), and (d) show the seasonal distributions of CTBT for summer, autumn, winter, and spring, respectively. The selection time of the CTBT ranges from 18:00 UT to 06:00 UT. The selection of this time range was due to the observation
time of the QMGWs and possible excitation time determined using ray tracing. The overall distribution of the CTBT for each season agrees with the propagation directions presented in panel (d) of Figure 3.

## 5 Case Studies

### 5.1 Case Study of July 20 - 21, 2017 Event

On July 20, 2017, around 22:00 UT, GW structures with $\lambda_H \sim 10$ - $60$ km were observed in the OH airglow propagation towards
the northwestern direction. The waves with similar wavelengths gradually propagated toward the west and southwest as time



progressed. In Figure 7, i) shows the $\lambda_H$ at each 1 hour whereas ii) shows the variation in $\phi$ at each hour. iii) shows the variation of $\phi$ in a polar plot representation. Panels a, b, and c of i) depict the variation of $\lambda_H$ between 30 - 40 km, 40 - 50 km, and 50 - 60 km of the GWs with time. The corresponding ii) and iii) of each panel represent the azimuth versus time and azimuth in a polar plot. The color representing $\lambda_H$ in i) at each hour corresponds to the same color in ii) and the arrow in iii).

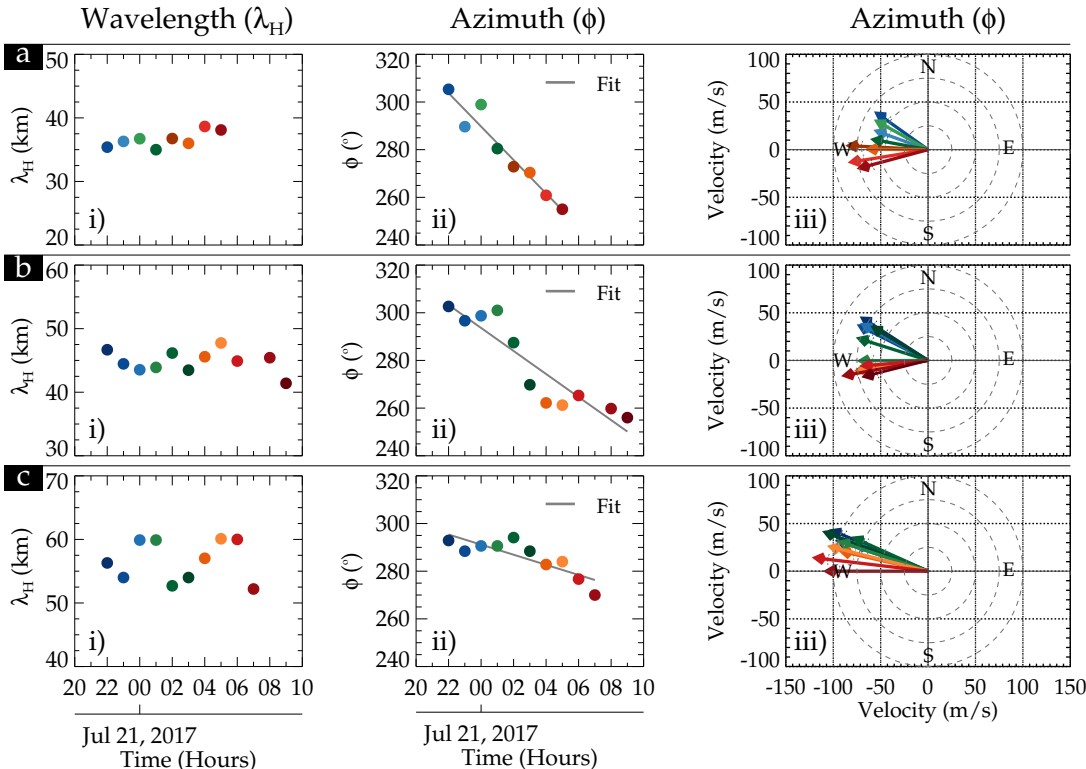

**Figure 7.** Propagation directions of 30 - 60 km - wavelength Gravity Waves. Panels i, ii, and iii are the $\lambda_H$, azimuth, and azimuth versus phase velocity in a polar plot.

We observed that the maximum variations in the three groupings of the wavelengths are within $\pm 5$ km. These variations fall within the average error range of each group. In relation to the variation in the azimuth with time, the change in the propagation direction is evident that all the groups of a wave propagated from the northwest at the beginning of the observation to the southwest at the end. The variations in the propagation direction in sub-panels ii) were affirmed in the polar plots in sub-panels iii).

### 5.1.1 Ray Tracing Result

Figure 8 shows the ray tracing results of the 29 QMGWs in the event observed on July 20 - 21, 2017. The color bar represents the number of each of the 29 waves. The hourly mean $\lambda_H$ of the three $\lambda_H$ groupings were ray traced from the OH emission



altitude. In Figure 8a, the horizontal line indicates 87 km, whereas the dot with similar colors to the model wind ray path of the wave is the reflection point. The squares in Figure 8b represent the position where the wave stops.

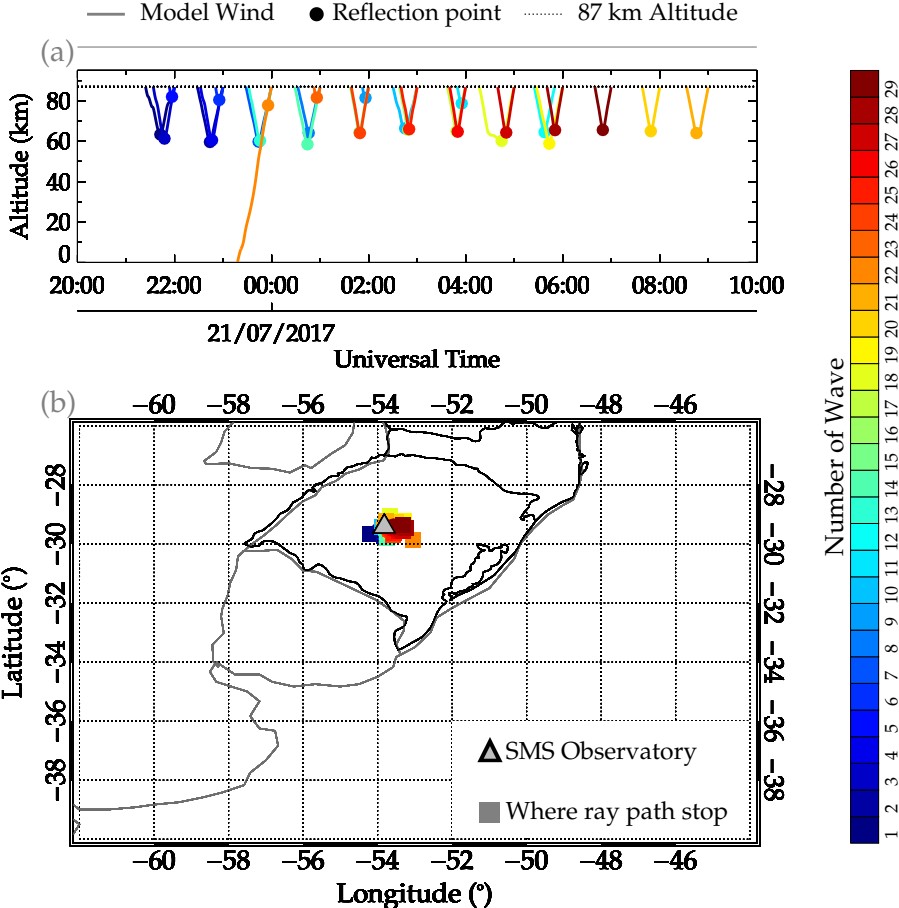

**Figure 8.** Ray tracing results of the Quasi-Monochromatic Gravity Waves on July 20 - 21, 2017.

Figure 8(a) shows that the ray tracing started at 87 km. However, we observed that almost all the wave reflected. Reflection of GWs occurs where and when the Brunt-Väisälä frequency (N) is nearly equal to the wave frequency ($\omega$). This result showed that except for Wave # 21 (see Figure 8a), which reached the troposphere, the others reflected between 60 and 85 km. The reflection of these waves suggests the possibility of an evanescence layer (m < 0). The subject of the ducting will be discussed in the following section. In panel (b), the position of the reflection in space is distributed around the observation site, showing

that the GWs did not travel far. Even Wave #21 propagated to the troposphere only 100 km from the observation site. The stopping point of this wave is not close to any convective system corresponding to the time when the ray path reached the tropopause.





### 5.1.2 Convective Sources

Figure 9 shows the minimum cloud top brightness temperature (CTBT) distribution in space between 18:00 UT on July 20,
2017 to 06:00 UT on July 21, 2017, and the vertical distribution of CTBT with overshooting top (OT) height.

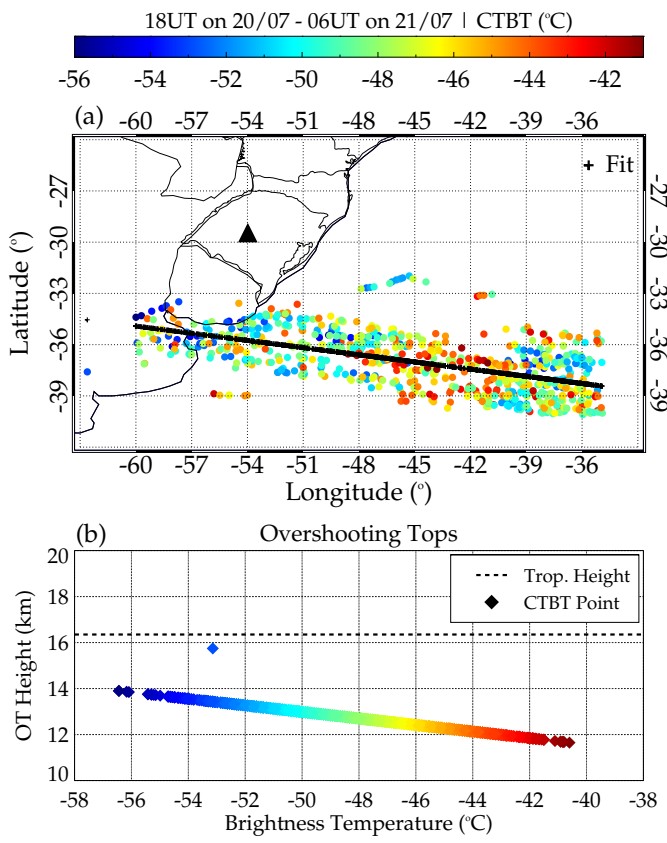

**Figure 9.** Distribution of (a) minimum cloud top brightness temperature (CTBT) in space and time, between 18:00 on July 20, 2017 to 06:00 on July 21, 2017, and (b) CTBT with OT height.

The minimum CTBT within longitude -63° to -33° and latitude -42° to -24° is determined for each 1° by 1° grid box from 18:00 UT on July 20, 2017 to 06:00 UT July 21, 2017 as shown in Figure 9(a). The composite plot of all the CTBTs is then plotted over the map to see their distribution relative to the stopping positions of the ray path. We observed in Figure 9 that the closest CTBT to the ray path stopping position is ~300 km for Wave #21, which reached the troposphere. It is important
to state that the other waves reached a minimum of 60 km altitude. Therefore, Figure 9(a) shows that these waves unlikely originated from the convective system. It is because no CTBT in Figure 9(a) did overshot, as shown in Figure 9(b).

The tropopause height obtained from a radiosonde sounding at Santa Maria (29.69°S, 53.27°W) on July 21 at 00:00 UT was ~16.35 km. The highest overshooting top within the time range considered was ~16 km. Several research works (e.g., Nyassor et al., 2021, and references therein) showed GWs can be generated through overshooting of the tropopause (mechanical



oscillator mechanism). However, for this mechanism to be feasible, the CTBT must be colder than the tropopause temperature signifying overshooting of the tropopause. This results therefore implies that overshooting of the tropopause is not the source mechanism of the waves observed on July 20 - 21, 2017. Fritts and Alexander (2003) mentioned that a convective system could also generate GWs through three mechanisms, latent heat, obstacle effect, and mechanical oscillator effect. Knowing that the mechanical oscillator effect is not responsible for generating the GWs on July 20 - 21, 2017, the other mechanism will be

explored later in the paper.

## 5.2  Case Study 2 - August 15 - 16 and 20 - 21, 2017 Events

Similar to the July 20 - 21, 2017 event, GW structures with $\lambda_H \sim$30 - 50 km were observed in the OH airglow images propagating toward the northwestern direction. Contrarily, these waves propagated mainly toward the northwest throughout the entire night. The variation in the $\lambda_H$ and $\phi$ at each 1 hour and the $\phi$ in a polar plot are presented in Figure 10. The sub-panels i), ii),

and iii) have the same meaning as defined in Figure 7. Panels (a) and (b) depict the subplot for i), ii), and iii) of $\lambda_H$ between 30 - 40 km and 40 - 50 km for the August 15 - 16 event, respectively, whereas panels (c) and (d) show that of the August 20 - 21 event.

In subpanel i) in panel (a), no significant variations were observed in the $\lambda_H$. The propagation direction ($\phi$) showed some variation with time but generally (the fit - blackline) varies from North to Northwest. Even though the 40 - 50 km GWs lasted

for just 3 h, it is clear that this GW was propagating mainly in the Northwestern direction (see subpanels ii) and iii) of panel (b)). The characteristics of these GWs clearly show that their source may be the same.

The GW structures observed on August 20 - 21, 2017, have a similar range of $\lambda_H$ as that of August 15 - 16, 2017. The two wavelength groups (30 - 40 and 40 - 50 km) variations in the $\lambda_H$ and the respective $\phi$ (in time and polar plot) are presented in panels (c) and (d) of Figure 10. Similarly to the propagation direction of the waves in Figures 9(a) and 9(b), the wave

(30 - 40 km) in Figure 10(c) propagates in a similar direction.



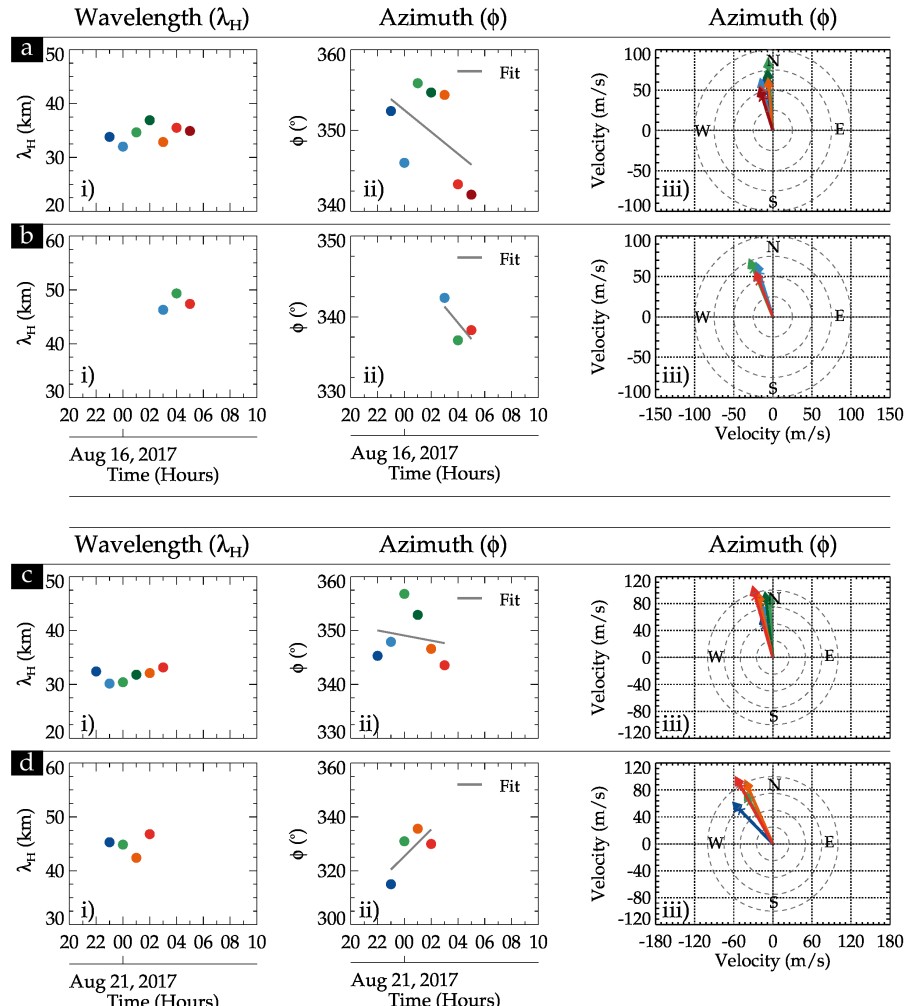

**Figure 10.** Similar to Figure 7 for only 30 - 50 km wavelength Gravity Waves. Panels i), ii), and iii) are the $\lambda_H$, azimuth, and azimuth in a polar plot. The case study of August 15 - 16, 2017, is presented in panels (a) and (b), whereas that of August 20 - 21, 2017 is presented in panels (c) and (d).

However, a 40 - 50 km wave propagated from the northwest to the North direction. The different propagation direction of the wave in this spectrum suggests that this wave might be excited by a different source. The propagation of these GWs is studied, and their possible source location is investigated using the ray tracing result presented in Figure 11.



### 5.2.1 Ray Tracing Results of August 15 - 16 and 20 - 21, 2017 GW Event

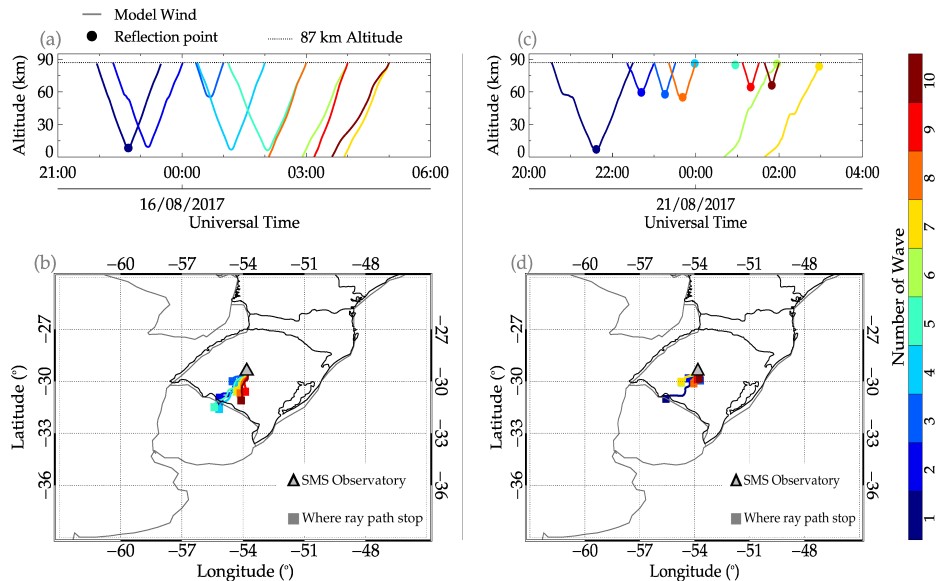

**Figure 11.** Ray tracing results of the Quasi-Monochromatic Gravity Waves on August 15 - 16, 2017, and August 20 - 21, 2017 events.

Figure 11 presents the ray tracing result of the August 15 - 16 and 20 - 21, 2017 QMGW events. Ten GWs were ray traced for each wave of the event. For the August 15 - 16 event, Figure 11(a) showed that Waves #6, #7, #8, #9, and #10 reflected between 50 and 60 km and then propagated to the ground. Waves #1, #2, #4, and #5 reflected around 10 km, whereas Wave #3 reflected at 60 km. The ray path of the wave in space (see Figure 11(b)), all the ray paths stopped at the Southwestern part of the SMS observatory. This indicates that the waves are most likely excited in the Southwestern part of Rio Grande do Sul or 235 the Northeastern part of Uruguay.

In Figure 11(c), we observed that all ten waves reflected at a point. Waves #4, #5, #6, and #7 reflected first at the OH emission altitude, among which Waves #6 and #7 propagated to the ground. Waves #4 and #5 could not propagate further upwards or downwards. Waves #2, #3, #8, #9, and #10 also reflected between 60 and 70 km. Wave #1, however, reflected ~5 km. As presented in Figure 11(d), the propagation of these waves in space showed that the waves are also generated in the 240 Southwestern part of the observation.

### 5.2.2 Convective Sources

Figure 12 shows the CTBT maps and the OT heights from 18:00 UT on August 15 to 06:00 UT on August 16 (panels a and b) and 18:00UT on August 20 to 06:00 UT on August 21(c and d) for case studies 2 and 3. The CTBT distribution in Figure 12(a) corresponds to the ray tracing result in Figure 11(b), whereas Figure 12(d) to that of Figure 11(c). From this plot (i.e., Figure 245 12), it has been observed that the distribution of the CTBT around the Southwest part of the observatory and the Northeastern



part of Uruguay agree with the stopping positions of the ray traced path. In particular, the ray tracing results in Figures 11(b) and 12(a) showed a clear correlation. Even though most of the ray paths in Figure 11(c) are reflected in the lower mesosphere, the ray paths that reached the troposphere agree with CTBT distribution. In both cases, the CTBT maps showed no strong convective activity. This is seen in the brightness temperature of the individual CTBT scales shown in the color bar.

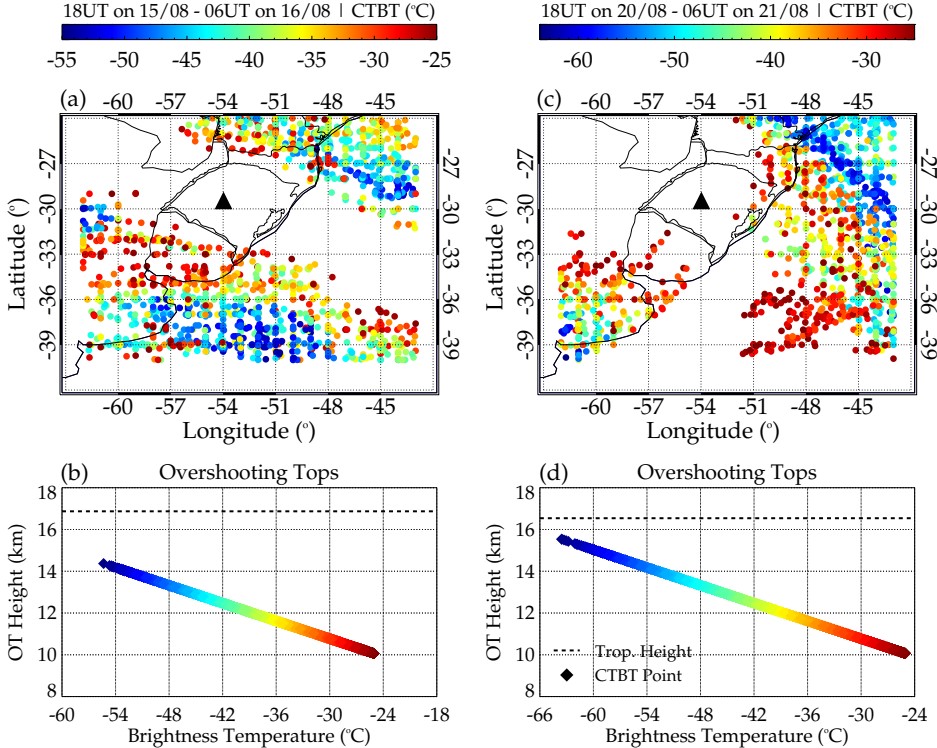

**Figure 12.** Distribution of (a) minimum cloud top brightness temperature (CTBT) in space between 18:00 on August 15, 2017, to 06:00 on August 16, 2017, and (b) CTBT with OT height. (c) CTBT in space between 18:00 on August 20, 2017, to 06:00 on August 21, 2017, and (d) CTBT with OT height

In Figure 12 (b and d), the individual OT heights are plotted as a function of brightness temperature. It can be observed that throughout the 12 hours, none of the CTBT/OT was higher than the tropopause height. This indicates that no overshooting by the convective system occurred; hence the mechanical oscillator effect of GW excitation cannot be the mechanism that excited these waves. However, other mechanisms can be the GW excitation mechanism of these waves. The general characteristics of the convective system during these nights showed characteristics of the activities of the cold front. Now, other mechanisms that

can excite the observed waves are investigated.





## 5.3 Lightning Distribution

Lightning activity is used to indicate the severity of deep convection. Nyassor et al. (2021, 2022) used lightning distribution in space to show the direct relationship to CTBT. Nyassor et al. (2021, and references therein) used the lightning rate as an indicator of overshooting of the tropopause while investigating the sources of three (3) concentric gravity events. Strong

correlations were observed between the lightning rate and overshooting tops in space and time. In this current study, the lightning activity in space (Figure 13) and time (not shown) were used to show whether the convective system present during the three case studies was active.

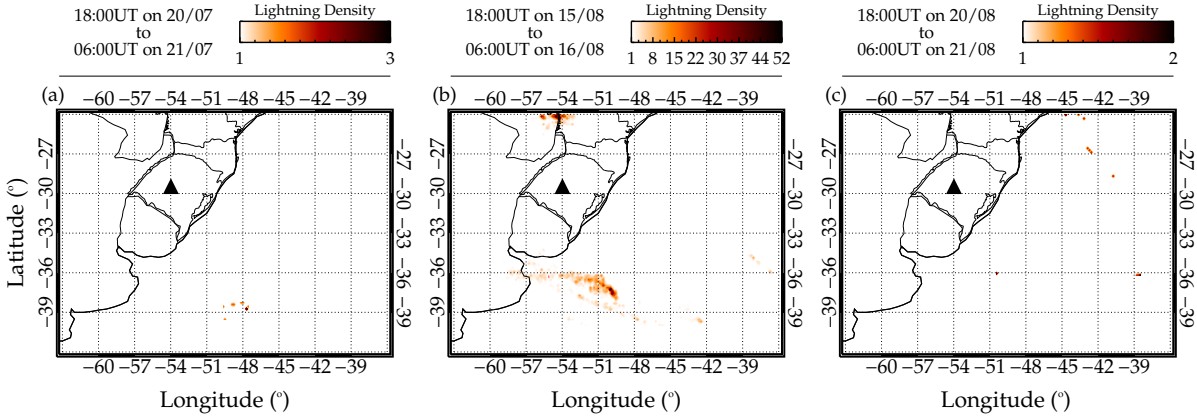

**Figure 13.** Lightning activities during the quasi-monochromatic gravity waves (QMGWs) case studies one, two, and three. The lightning activity distribution during case study one is shown in (a), for case study two in panel (b), and in panel (c) is that of case study three.

To determine the lightning density, we binned the lightning flashes by $0.15° \times 0.15°$ in longitude and latitude (Nyassor et al., 2021, 2022) from 18:00 UT to 06:00 UT. The density distribution was then overplotted on the map, as shown in Figure

13. Interestingly, the distribution density of the lightning during these case studies is low, especially for the cases of July 20 - 21 and August 20 - 21. The maximum density distribution is event 2 (Figure 13b). However, the distribution of this event is far from the ray traced source locations. This is another indication that these convective systems were not active. Even though the lightning distribution of the August 15 - 16 QMGW event (Figure 13b) was relatively high, the lightning rate (not shown) did not present characteristics of overshooting. Using lightning activity, it has been further proven that the observed waves are not

excited through the overshooting mechanism.

## 6 Background conditions on the propagation of the case studies

The propagation of GWs is controlled by the atmospheric background field, especially wind, and temperature. The state of the wind and temperature determine whether a wave is propagating or evanescence (ducted or trapped) (Gossard and Hooke, 1975). Ducted GWs can propagate for long distances without losing energy (Bageston et al., 2009). Bageston et al. (2009) and





Fechine et al. (2009) showed that ducted waves due to either thermal or Doppler duct enhance the longer horizontal propagation
of GWs over a long duration. Thermal ducts are formed when there is a temperature inversion layer, whereas the Doppler duct
is formed when a wind shear exists. A dual duct is formed when both temperature inversion layer and wind shear exist at the
same altitude (e.g., Chimonas and Hines, 1986; Isler et al., 1997; Nappo, 2013; Walterscheid et al., 1999).

During these three case studies, temperature profiles obtained from SABER sounding showed an inversion layer within 60
to 90 km. The case studies of July 20 - 21, August 15 - 16, and August 20 - 21 have a mesospheric inversion. Also, vertical shear
was present in the zonal wind. Therefore, the ducting condition was determined by utilizing the SABER temperature profile
and the concatenated wind profiles obtained from MERRA-2 and HWM14.

In Figure 14, the background analysis of the propagation characteristics of the July 20 - 21, 2017 QMGW event is presented.
The SABER temperature profile and its corresponding potential temperature are presented in panel (a). The Brunt Väisälä
frequency profile was estimated using (Fritts and Alexander, 2003)

$$N = \sqrt{-\frac{g}{\theta}\frac{d\theta}{dz}}, \tag{1}$$

which is presented in panel (b), with $\theta$ being potential temperature, $g$ the gravitational acceleration, and $z$ the altitude. In
panel (c), the wind in the direction of the wave for propagation directions of 315°, 290°, and 270° is shown, whereas the profile
of the vertical wavenumber (m) adapted from Vadas and Fritts (2005) is shown in panel (d).

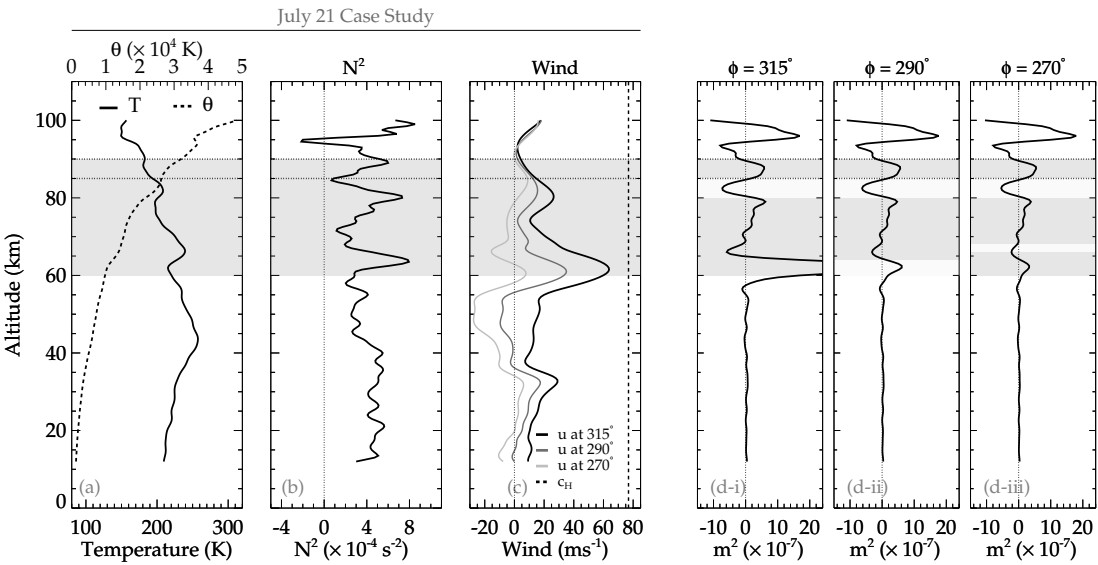

**Figure 14.** Propagation characteristics during the July 20 - 21, 2017 QMGWs case study.

Panel (d) has three subpanels: i, ii, and iii. These subpanels represent the directions of propagation of the wave. As seen
earlier in Figure 7, the wave in this case study propagated from northwest to southwest. As a result, propagation directions in
315°, 290°, and 270° are considered to verify if a duct exists in all directions during the propagation of the wave.





From the analysis in Figure 14, it is seen that there exist a duct in all three propagation direction considered. The existence of the duct implies that the background creates the necessary condition favourable for the wave to propagate in this region for a long distance and time. During this QMGW event, the propagation of the observed wave for about nine (9) hours with quasi-horizontal wavelength suggests: (a) a possible propagation in a duct and (b) a source emitting GWs at a constant spatial and temporal scale over a long time. Regarding the longer propagation, the presence of the duct affirms the longer propagation of these QMGWs with similar spatial characteristics from the beginning to the end of the observation. Various researchers have used ducts (e.g., Xu et al., 2015) to explain the longer propagation of the wave reported in their work. Similar to the result of Xu et al. (2015) and references therein, it can be concluded that the July 20 - 21, 2017 event was ducted hence the longer propagation over such a long time.

A similar analysis is conducted for the August 15 - 16 and August 20 - 21 QMGW cases, as shown in Figure 15. The profile of the parameters in panels (a), (b), and (c) are similar to that of Figure 14 except for panel (d), where $m^2$ of only a single propagation direction obtained in panel (c) is presented. In both case studies, the $m^2$ profile showed two ducts between 75 - 95 km altitude ranges. The ducts of these two cases are not precisely within the peak of OH emission layer altitude (i.e., 87 km) and are narrower than that of July 20 - 21, 2017. However, these ducts can support longer horizontal propagation of the observed QMGWs.







**Figure 15.** Propagation characteristics during the August 15 - 16 and August 20 - 21, 2017 QMGWs case studies.

In Figure 11, the ray tracing result of the August 15 - 16 case studies (panels (a) and (b)), except for one of the wave (i.e., wave #3), the remaining nine (9) reached the troposphere. This indicates that these waves were excited in the lower atmosphere and in the southwest of the observation site, with waves #6, #7, #8, #9, and #10 first reflecting around 60 km. Similarly, in panels (a) and (b), the ray tracing result showed that only three waves reached the troposphere. The remaining waves reflected above 60 km of altitude. These propagation characteristics, however, indicate the condition, that is, $c_H = N/k + U$, for reflection, is satisfied here Heale and Snively (2015).





The ray tracing results for the three case studies could not capture the trapping of the waves but could only capture the reflection because the phase front of the wave and the background wind are the same. Also, the zonal component of the wind during these events peaked within 60 to 70 km. In a simulation study conducted by Heale and Snively (2015), for small-scale gravity waves (SSGWs), they found that their simulated wave ray path reflected along the largest magnitude negative phase front of the background wind. All this evidence showed that most of the observed QMGWs are ducted, allowing longer propagation. Next, we investigate the source and related mechanism that emitted the spectrum of waves observed in the mesosphere.

## 7 Other Wave Sources and Source Mechanisms

### 7.1 Cold Fronts

A cold front is the leading edge of a cooler air mass at ground level that replaces a warmer air mass and lies within a pronounced surface trough of low pressure. A cold front generates a cumulous cloud with precipitation, emitting GWs. Since the systems for case studies 1, 2, and 3 are not overshooting, further analysis of the characteristics of the system is conducted using GOES images to study the cold front characteristics. According to Schmit et al. (2017), among the GOES-16 products, channel 10 captures lower/midlevel water vapor (fronts) activities between 500 and 750 hPa (2.5 - 5.5 km) in the infrared wavelength at 7.3 $\mu$m.

Figure 16 presents a three-hour resolution time series of GOES-16 infrared images of a cold front between 18:00 UT and 06:00 UT. Figure 16(a-d) presents the spatiotemporal evolution of the cold front during the July 20 - 21, 20217 case study, whereas panels (e-h) and (i-l) of Figure 16 present that of August 15 - 16 and 20 - 21 case studies, respectively. The time of each image is written in the upper left corner. The color bar at the upper left shows the temperature scale of the cold front. Cold fronts are used, among others, to monitor severe weather potential. For this reason, complementary data such as reanalysis and space-borne observation were used to investigate the possibility of severe weather.

First, convective available potential energy (CAPE), an essential parameter in predicting severe weather, is used. From CAPE, the maximum updraft velocity ($w = \sqrt{2.CAPE}$) is mostly used to determine possible overshooting of the tropopause that can lead to gravity wave excitation. As discussed earlier, no overshooting was observed before the observation of these case studies. However, to confirm that no overshooting was observed, CAPE maps within the same time and spatial range, as shown in Figure 16, were plotted and presented in Figure 17. The color bar in the upper left corner shows the values of CAPE.

In Figures 16 and 17, the contour lines of omega ($dp/dz$) at 850 hPa were overplotted on the cold front and CAPE maps. In Figure 16, the contour lines and their magnitude are represented in red, whereas in Figure 17, they are in gray lines. The omega data were obtained from the National Centers for Environmental Prediction (NCEP). More details on the omega data can be seen elsewhere in Kanamitsu et al. (2002). The following section will discuss the omega contours, the cold front, and CAPE. The omega is plotted over the cold front and CAPE maps to relate the regions with cold brightness temperature and high CAPE values to strong upward vertical air motion. The omega (upward vertical motion of air mass) in gray contour lines are overplotted on the spatio-temporal evolution of (CAPE) maps. Even though in Figures 16 and 17, omega at 850 hPa was overplotted on the cold front and CAPE maps, the question still remains: what are the characteristics of omega with altitude?







**Figure 16.** Spatio-temporal evolution of cold front in GOES-16 channel 10, 7.3 μm with the omega (upward vertical motion of air mass) overplotted in red contour lines.





**Figure 17.** Spatio-temporal evolution of convective available potential energy (CAPE) with the omega (upward vertical motion of air mass) overplotted in red contour lines.





Figures 16 and 17 present results of coincident observations and reanalysis data used to investigate the state of the tropospheric activity. Figure 16 clearly shows the cold front passage during the three (3) selected case studies. For the case of July 20‑21, the cold front was moving eastward, whereas August 15‑16 and August 20‑21 were moving southwestward. A close observation at the omega over the cold front showed that the region with colder temperature has negative omega at 850 hPa, which indicates a consistent ascending motion of the atmosphere (Xu et al., 2015). In the cases of July 20‑21 and August 15‑16, regions with negative omega are over Uruguay and part of Argentina between 18:00 UT and 06:00 UT. These regions are to the south and southwest of the observation site. For the case of August 20‑21, the region of the passage of the cold front was over latitudes higher the -25° and the majority over the Atlantic Ocean. The omega with a negative sign coincides with these regions. The characteristics of the cold fronts are further affirmed using the CAPE maps (Figure 17).

CAPE is used as an indicator of atmospheric instability, which measures the integrated work that the upward buoyancy force would perform on a given mass of air to rise vertically through the entire atmosphere (Holton, 1992). In this work, we used the CAPE further to show the state of instability of the atmosphere. Several works by researchers (e.g., Vadas et al., 2009; Xu et al., 2015; Nyassor et al., 2021) used CAPE (updraft) to infer the possibility of severe weather that can lead to overshooting and consequently GWs excitation. According to the Storm Prediction Center of National Oceanic and Atmospheric Administration (Nyassor et al., 2021), CAPE is classified as marginally unstable when $0 \leq \text{CAPE} \leq 1000$, moderately unstable when $1000 \leq \text{CAPE} \leq 2500$, very unstable when $2500 \leq \text{CAPE} \leq 4000$, and extremely unstable when $\text{CAPE} \leq 4000$.

The higher the value of CAPE, the greater the possibility of the formation of severe weather and also the higher the maximum updraft velocity that may lead to overshooting of the tropopause and thereby exciting GWs. In the case studies considered in this work, the values of the CAPE were very low, especially in the regions indicated by the ray tracing to be the possible source location of the QMGWs, as shown in Figure 17. This, therefore, strengthens the result in Figures 9 and 12 that the source mechanism of the GWs observed in these case studies is not through the mechanical oscillator (overshooting) effect.

The observed CTBT map did not show overshooting implying that the clouds did not extend too high to the upper troposphere. To further confirm this, the vertical column of the cloud (see Figure A1 in Appendix A) was analysed. For the case study of July 20‑21, there was no observation of CloudSat. On the other hand, there were observations during the case studies of August 15‑16 and August 20‑21, where CloudSat passed right through the region of negative omega (see Figures 16 and 17). Clearly, Figure A1 showed the presence of only low-level clouds. Another piece of evidence to show that the three case studies in this work were not excited through the mechanical oscillator (overshooting) mechanism is the vertical profile of omega at fixed longitude and varying latitude as shown in Figure A2. The vertical profiles of the cloud and omega, are specifically presented to affirm further the result of the cold front (Figure 16) and CAPE (Figure 17) maps that no overshooting took place despite being clearly depicted in Figure 9 and 12. For details on the vertical profiles of the clouds and omega, see Appendix A.

All this evidence, clearly shows that the QMGW events selected for the three case studies are not excited through the mechanical oscillator effect mechanism of convection. However, other mechanisms associated with a cold front can excite these waves. We now investigate this possible mechanism.





## 7.2 Wind Shear

Cold fronts are known to be characterized by temperature field but also by pressure, wind speed, and direction that precede and succeed its passage. Pressure zones, wind speed, and direction can also identify cold fronts. The characteristics of the wind are such that a sudden change in wind direction commonly occurs with the passage of a cold front. According to Van Den Broeke (2022), before the arrival of the front, winds ahead of the front (in the warmer air mass) are typically out of the south-southwest. Still, the winds usually shift to the west-northwest (in the colder air mass) after the front passage. These case studies, however, occurred during the winter season, when strong wind shear and jet streams are prominent.

Strong wind shears in the upper troposphere-lower stratosphere are responsible for generating tropopause shear layers, which generate local turbulence and consequently can lead to mixing air between these two different layers (Kaluza et al., 2021). This mixing air contributes to the emergence of dynamic instabilities that conduct waves to overturn, followed by the turbulent flow breakdown in this transition region. This approach has been discussed in the context of clear-air turbulence (CAT) since 1970 (e.g., Shapiro, 1976, 1978). Recently, a midlatitude cyclone was simulated using the high-resolution numerical model, in which many turbulences were reported. This information highlights the importance of the tropospheric jet streak, wind speed, and shear enhancement within upper-tropospheric outflow with the occurrence of CAT and the generation of gravity waves on different scales (Trier et al., 2020). A jet streak is a section of the overall jet stream in which winds are greater along the jet core flow than in other parts of the jet stream.

Following this approach, the horizontal winds at 200 hPa are analyzed for each event in these selected case studies (Figure 18). The horizontal wind speed (contour plot) and direction (overplotted vector in red arrows) at 200 hPa of the case studies of 20 - 21 July, 15 - 16 August, and 20 - 21 August are presented in panels (a), (b), and (c) of Figure 18, respectively. These winds are obtained from the National Centers for Environmental Prediction (NCEP) (Kalnay et al., 1996). The subpanels (i) and (ii) represent the wind speeds and directions at 18:00 UT on the previous day and 06:00 UT on the next day of each case study with their respective speed in the color bar.

It is important to note that these events occurred during the winter when the polar jet stream (wind ~60 m/s) is generally displaced southward, as observed in these three events. Bertin et al. (1978) showed many possible source mechanisms of gravity waves observed in the mesosphere that appear to be closely related to tropospheric jet streams, principally on the polar side of jets. Also, Mastrantonio et al. (1976) showed that the gravity waves generated by tropospheric jet streams may have the ability to propagate vertically to the upper atmosphere, such as the ionosphere (Mastrantonio et al., 1976).





**Figure 18.** Horizontal wind at 18:00 UT and 06:00 UT on 20 - 21 July 2017 (a(i - ii)), 15 - 16 August 2017 (b (i - ii)), and 20 - 21 August 2017 (c (i - ii)).

Figure 18 (a) shows a strong and clear bifurcation of the strong wind flow close to longitude -60°, coinciding with the source location of the gravity waves. This bifurcation was persistent with strong wind flow throughout the 12 hours, suggesting a
constant emission of the gravity waves in this region. It can be observed that the wind was toward the northeast direction at





18:00 UT on July 20 and 06:00 UT on July 21. The propagation direction of the wave during this case study was northeastward at the beginning of the observation on July 20, 2017.

Now we investigate the source of the QMGWs of the 15 - 16 August case study. Similar to the case study of 20 - 21 July, Figure 18(b) shows a confluence of the strong wind flow from the north and southwest towards the southeast direction over the region. This unidirectional wind flow may suggest a persistent and unidirectional emission of gravity waves throughout the 12 hours. Close to the observation site, the omega was upward where the clouds were formed, see Figure 16(e-h). In Figure A2(b), omega extends almost throughout the altitude ranges considered in panels i and ii.

Finally, similar to Figure 18(b), a confluence of the strong wind flow from the northwest and south to the southeast direction close to the region of study. The wind considered to be associated with the source mechanism of the case study of the 20 - 21 August QMGWs presented a different characteristic. Figure 10(c - d) shows that the two wavelength groups of these QMGWs have different propagation directions. The 30 - 40 km and 40 - 50 km wavelengths had no well-defined propagation direction. The ray tracing, on the other hand, showed that only three of the waves were generated in the troposphere. The remaining waves reflected above ∼60 km. The ray path of the wave that reached the troposphere revealed that these waves were generated in the southwestern part of the observation. In the troposphere, Figure 16(i - l) showed that the cold front extends from the northeastern, eastern, and southwestern parts of the observation site. This system is quite distant from the observation. Considering the propagation direction of the waves, there is no way these waves can be excited by this system. According to Pramitha et al. (2016), wind shear can excite GWs, so considering the propagation of this wave, wind shear is most likely the source of this wave.

The vertical profiles of the omega (Figure A2), zonal wind (Figure 19), and wind shear (Figure 20) at fixed longitudes are analyzed to identify the main characteristics of the vertical position of the jet streams close to the observation site. The fixed longitudes (Figures A2 and 19) in case study one are 60°W and 65°W (panels a), the case study two are 50°W and 60°W (panels b), and in the case study three are 50°W and 40°W (panels c), respectively. The fixed longitudes for Figure 20 are 62.5°W (case study one - panel a), 55°W (case study two - panel b), and 45°W (case study three - panel c), respectively.

The vertical positions of the jet streams are close to 400 hPa for these three case studies due to their occurrences during the winter season. The bifurcation of the wind flow is easily identified in panel (a) for the first case study, while the confluences of the wind flows are observed in case studies two (panels b) and three (panels c). The latitude with strong jet stream signatures corresponds to the latitude of the observation site.



**Figure 19.** Vertical zonal wind profile at fixed longitude stated in the upper left corner of subpanels (i) and (ii), and varying latitudes are presented. Subpanels (i), show the NCEP zonal wind profile at 18:00 UT on (a) July 20, (b) August 15, and August 20. In (ii), the profile at 06:00 UT on (a) July 21, August 16, and August 21 are shown.



The vertical profiles of the omega (Figure A2) show the ascendant motions slightly below the jet stream and descendant motions in the jet stream core, suggesting that the presence of the wind shear is close to the source region of all the events in
this study. This behavior of the vertical motions may trigger the physical processes responsible for generating the turbulence close to the tropopause, which can lead to gravity wave generation. The excited waves can propagate vertically to the upper atmosphere (see Mastrantonio et al. (1976) and Bertin et al. (1978)). Also, the vertical profile of the wind shear presented in Figure 20, estimated using horizontal wind obtained from the NCEP global forecast system (GFS) data, is studied.

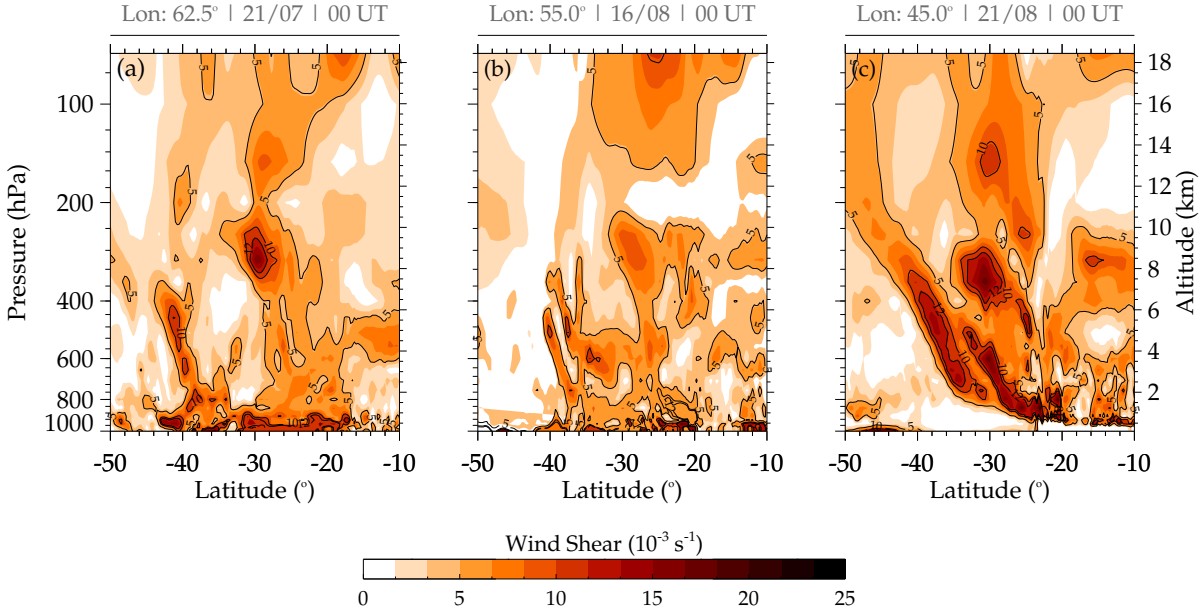

**Figure 20.** Vertical profile of vertical wind shear at 00:00 UT on 20‑21 July 2017 (a), 15‑16 August 2017 (b), and 20‑21 August 2017 (c). The color bar shows the scale of the wind shear.

The vertical profiles of the wind shear in Figure 20 show values greater than $15\times10^{-3}\text{s}^{-1}$ close to the center of the jet
streams (red boxes), where the vertical displacement of the wind shear (blue rows) is observed in these three events. This indicates the occurrence of turbulence close to the jet stream region with the vertical upward extension. These values are in accordance with literature which indicates the occurrence of CAT in the troposphere (Menegardo-Souza et al., 2022). The vertical extension of wind shear can generate gravity waves capable of propagating vertically.

Several possible source mechanisms for generating the selected case studies have been presented in the previous section.
However, some of these mechanisms cannot be responsible for the excitation of these events because they did not meet the necessary requisite conditions. For instance, the overshooting mechanism is not possible based on the inability of the convective system to overshoot. This is not obvious since the selected cases were observed in winter, where deep convection is less prominent. On the other hand, jet streams associated with the cold front have been strong before, during, and after these events.



Jet streams are relatively narrow bands of strong wind blowing from west to east in the upper troposphere and lower strato-
sphere (UTLS). As jet stream changes in intensity and location, the strength and motion of air masses are affected, and when
the air masses converge, they form fronts. When colder air mass replaces warmer air mass, colder fronts are formed. This is
the exact condition in the lower atmosphere during these events. As a result, further analysis was conducted on the jet stream
to establish a relationship between the jet streams - GW excitation mechanism and the observed GWs.

GWs observed in the upper mesosphere can be excited by jet streams in the lower atmosphere (Song, 2021, and references
therein.). From the above analyses, it is clear that the activities of jet streams may be the mechanism that led to the emission of
the observed QMGWs in the selected case studies.

## 8   Conclusions

In this study, 209 QMGWs were observed from April 2017 to April 2022, among which ray tracing results showed that 186
were excited in the troposphere, whereas the remaining 26 were reflected above. Statistically, it was observed that there was
a high occurrence of QMGWs in August, followed by July, with the least occurrence in May. Estimates of wave parameters
after applying spectral analysis revealed that the horizontal wavelength ranges between 10 and 55 km with an average value of
22.50 km, periods between 0 and 80 min, and phase speeds between 0 and 100 m/s.

The propagation direction of the waves showed quite anisotropic distribution with dominant distribution within northeast
through north to northwest and east to south. These propagation directions are consistent with the ray traced source locations
and the CTBT distributions. Relating the source locations to the CTBT locations, it was observed that most of the waves were
not excited by convection activity, as revealed by the seasonal distribution of CTBT. The duration of the QMGWs in OH images
lasted between 2 to 10 hours with the 2 hours duration having the highest number of QMGW events, whereas the 10 hours had
the least QMGW events. The propagation time of the waves from the OH emission layer altitude to the troposphere ranges from
0 to 9 hours. Besides the total QMGW cases presented, three QMGW events on 20 - 21 July 2017, 15 - 16 August 2017, and
20 - 21 August 2017 were selected for case studies. The selected waves were grouped according to their horizontal wavelengths,
after which their propagation dynamics were studied relative to their source.

The propagation directions of the case study of 20 - 21 July 2017 QMGWs showed that the directions of the waves varied
from the northwest through north to southwest. However, the ray tracing result showed that except for one wave that reached
the troposphere, the rest reflected above ∼60 km. This is an indication of the possibility of ducting or reflection. To further
investigate the details of this possibility, propagation characteristics due to the background field were conducted. It was found
that the duct enhanced the longer propagation of this event and also the changing propagation direction. The source of these
case studies was most likely jet streams. Similarly, the source of the 15 - 16 August and 20 - 21 August case studies is also most
possibly due to jet stream, and one QMGW event was also ducted. Contrarily, most of the ray paths of these waves reached
the troposphere, signifying these waves were excited in the troposphere. In the case of the 20 - 21 August case studies, about 7
waves reflected above 60 km.



In conclusion, the current study presents statistical evidence of the occurrence of QMGWs. Their occurrences were further investigated in detail using the seasonal distribution of the propagation directions in relation to the seasonal CTBT distributions in space indicated by the ray tracing to be the possible source location. Due to the peculiar characteristics of the 3 case studies and their occurrence in the winter month, they were chosen for further detailed studies. These case studies were ducted; as

a result, they could propagate longer distances with quasi-horizontal wavelength for a long time. The sources of these case studies were not related to convective activity but to jet streams.

*Data availability.* The airglow data used to produce the results of this paper were obtained from the Southern Space Observatory at São Martinho da Serra, which is supported by the Southern Space Coordination of the National Institute for Space Research. The airglow data are available on the web page of the "Estudo e Monitoramento Brasileiro do Clima Espacial" (EMBRACE/INPE) at http://www2.inpe.br/climaespacial/

portal/en (EMBRACE, 2022). The GOES-16 cloud top brightness temperature (CTBT) maps were provided by the Center for Weather Forecasting and Climate Studies (CPTEC/INPE) and are available at http://satelite.cptec.inpe.br/ CPTEC (2023). The radiosonde data were provided by the University of Wyoming and can be accessed through http://weather.uwyo.edu/upperair/sounding.html (UWYO, 2022). ERA5 data can be accessed from the Copernicus Climate Data Store at https://cds.climate.copernicus.eu/ (Hersbach et al., 2018), whereas MERRA2 can be accessed through https://doi.org/10.5067/WWQSXQ8IVFW8 (GMAO, 2015). NCEP-NCAR Reanalysis 1 data provided

by the NOAA PSL, Boulder, Colorado, USA, from their website at https://psl.noaa.gov.

## Appendix A

### A1 CloudSat 2B-GEOPROF vertical profile of the cloud

The level 2B GEOPROF R04 and R05 products of CloudSat determine levels in the vertical column sampled by CloudSat that contain significant radar echo from hydrometeors and then provides the radar reflectivity factor. GEOPROF also includes a

product that estimates the expected gaseous absorption loss for the observed reflectivity, which is dependent on water vapor fields from the European Centre for Medium-Range Weather Forecasts (ECMWF). The Moderate Resolution Imaging Spectroradiometer (MODIS) cloud fraction from MOD35 associated with the radar surface footprint and several other flags indicate the homogeneity of the MODIS data and the quality of the CloudSat data. Details on the GEOPROF algorithms and structure of the HDF-EOS output files can be found in Marchand et al. (2008) and the Level 2 GEOPROF Product Process Description

and Interface Control Document.

Using the level 2B GEOPROF R04 and R05, the vertical column of the clouds and the CloudSat track were obtained. In Figure A1, the track of the satellite and the cloud vertical column are shown for the cases of August 15 - 16 (panel a) and August 20 - 21, 2017 (panels b and c). The satellite did not pass during the July 20 - 21 event; hence no plot is presented for this day. In Figure A1, subpanel (i) represents the track of the CloudSat, while subpanel (ii) shows the vertical column of the

cloud with time. In panel (ii), the horizontal dashed lines depict the radiosonde tropopause altitude at 00:00 UT on August 16 and 21. The color bar shows the scale of the radar reflectivity factor (dBZ).





**Figure A1.** CloudSat 2B-GEOPROF vertical profile of the cloud during (a) August 15 - 15, 2017, and (b and c) August 20 - 21, 2017 QMGWs case studies. (i) is the satellite track, and (ii) the vertical profile of the cloud.

Between 17:33 to 17:38 UT on August 15 (Figure 16b) the vertical column of cloud for the nearby track of sounding is presented. This time is earlier than the time interval for the cold front and CAPE maps. However, the figure is presented to prove the existence of the cloud, but only a low-level cloud between 17:35 and 17:38 UT was observed.

Before the case study of August 20 - 21, CloudSat made two passages, one between 16:16 - 16:20 UT (Figure A1b(ii)), with the track shown in Figure A1b(i). The satellite passed through the location of the cold front, which also corresponded to the





negative omega region. A well-defined profile of the vertical column of the cloud was captured since the satellite passes right through the middle of the cloud within this time range. Even though the CTBT maps showed cold cloud top temperatures (see panels i, j, k, l of Figure 14), the vertical column of clouds extended up to about 12 km (see Figure A1), about 4 km lower than

the tropopause height. This, further, shows that no overshooting occurred; hence this source mechanism cannot be responsible for generating the GWs of this case study. The second satellite track shown in Figure A1c(i) between 17:52 and 17:59 UT could not capture any cloud profile (see Figure A1c(ii)) because there was no cloud present at that time. At 18:00 UT, a system of clouds was seen progressing from the southwestern part towards the northeast but dissipated as time progressed. Without clouds near the source location of the QMGWs of this case study (Figure 11d) implies other mechanisms will be the source of

this case study.

## A2  Vertical profile of omega ($\omega$)

Figure A2 presents the vertical profiles of omega for the case studies of July 20 - 21, August 15 - 16, and August 20 - 21. In panels (a), (b), and (c), the plot of omega with altitude (hPa and km) for the three case studies is presented. The omega used in these plots were obtained at varying latitudes of -50 to -10°S and fixed longitudes; at 60°W (i) and 65°W (ii) for panel (a), 50°W (i), and 60°W (ii) for panel (b), and 50°W (i) and 40°W (ii) in panel (c). The altitude in kilometers for all panels

corresponding to the pressure levels in hPa (on the left side of (i)) is on the right side of (iii).

From the omega vertical profiles of the three case studies at a fixed longitude and varying latitudes, we observed negative omega in the July 20 - 21 event, extending from 0 to the tropopause for panel a(i) of Figure A2. For panel a(ii), at 65.00°W, -50 to -10°S, the negative omega extended almost throughout the entire pressure/altitude range considered. The omega ascended

almost throughout the profile for the 15 - 16 August case study. However, a descending region from 200 to 600 hPa and -35 - 25°S in panel b(i) and -45 - 30°S in panel b(ii). A different omega distribution was observed in the case of August 20 - 21 (i.e., Figure A2). The positive omega (signifying downward motion of air) dominates the altitude range from 3 to ~15 km and between -50 and -20°S. Beyond -20°S omega was ascending. Figure A2 shows that, even with the upward movement of the air mass, its effect did not lead to the formation of clouds and eventually to the excitation of GWs. The omega vertical profile

further affirms the previous evidence that these three QMGW events are not excited through overshooting. However, omega is used not only as an indicator for cloud formation but also for vertical wind shear.







**Figure A2.** Vertical profile of omega ($\omega$) at 18:00 UT, 00:00 UT and 06:00 UT on 20 - 21 July, 2017 (a (i - iii)), 15 - 16 August, 2017 (b (i - iii)) and 20 - 21 August, 2017 (c (i - iii)).



*Author contributions.* CMW wrote the article and performed most of the analysis. PKN assisted in the development and validation of the methodologies and in the revision of the manuscript. LAS assisted in the development and validation of the meteorological methodologies and in the revision of the manuscript. CAOBF assisted in the development and validation of some of the methodologies and the revision of the manuscript. JVB provided the all-sky images. KPN provided lightning data and revised the manuscript. HT revised the manuscript, and DB helped in the development and validation of some of the methodologies and the revision of the manuscript. DG revised the manuscript.

*Competing interests.* The contact author has declared that none of the authors has any competing interests.

*Acknowledgements.* Cristiano M. Wrasse thanks the Coordenação de Aperfeiçoamento de Pessoal de Nível Superior (CAPES) and the Conselho Nacional de Desenvolvimento Científico e Tecnológico (CNPq) for the support. Thanks are given to the Brazilian Ministry of Science, Technology and Innovations (MCTI) and the Brazilian Space Agency (AEB). Prosper K. Nyassor, Cosme A. O. B. Figueiredo and Diego Barros acknowledge the support of Fundação de Amparo à Pesquisa do Estado de São Paulo (FAPESP). Also, Cosme A. O. B. Figueiredo acknowledges Fundação de apoio à pesquisa do estado da Paraíba. Ligia A. da Silva grateful for financial support from China-Brazil Joint Laboratory for Space Weather (CBJLSW), National Space Science Center (NSSC) and the Chinese Academy of Science (CAS). The authors thank the Estudo e Monitoramento Brasileiro do Clima Espacial (EMBRACE/INPE) for the provision of all-sky data and the Center for Weather Forecasting and Climate Studies (CPTEC/INPE) for the cloud top brightness temperature (CTBT) maps. Also, we appreciate the Brazilian Lightning Detection Network (BrasilDAT) from the Earth Sciences Department (DIIAV/CGCT/INPE) supported by EarthNetworks for the lightning data, the Department of Atmospheric Science of the University of Wyoming for providing the radiosonde data, and National Centers for Environmental Prediction for the tropospheric data and the Global Forecast System (GFS) Model.





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
