# Peer review of "Studies on the Propagation Dynamics and Source Mechanism of Quasi-Monochromatic Gravity Waves Observed over São Martinho da Serra (29°S, 53°W), Brazil"

_EGUsphere, 2023_

## Author Comment (AC1)

**Referee #2 Evaluations:**

**Reviewer #2 (Formal Review for Author):**

**General comments:**

The paper considers about two hundred events of quasi-monochromatic atmospheric gravity waves (QMGWs) which were acquired over five years of observations in Southern Brazil, obtained with an OH all-sky imager. Gravity waves exhibited a seasonal dependence on the propagation direction with an anisotropic behavior. Typical characteristics of QMGWs are horizontal wavelengths of $10\text{-}55\,\text{km}$, observed periods of $5-74$ minutes and observed phase speeds up to $100\,\text{m/s}$. A complimentary backward ray tracing analysis allowed the authors to reveal potential sources of QMGWs. I have found the paper to be interesting to the atmospheric community. At the same time, there are issues that are needed to explain and clarify. This is why a revision of the present paper is needed.

**Specific comments**

The authors would like to thank the reference for taking time off to review this manuscript and for the comments, suggestions, and corrections. We have given the response accordingly.

- **Lines 2 - 3:** "The observations were made using OH all-sky imagers hosted by the Southern Space Observatory (SSO)..."

  **Question:** How many all-sky imagers were used? One or several?

  **Response:** Gravity wave observations were taken at the Southern Space Observatory (SSO), located in São Martinho da Serra (SMS) (29.44°S; 53.85°W), Rio Grande do Sul, Brazil, using a single-filter (OH) imager that began operation from April 2017 till date.

  Based on this comment, the entire Section on OH All-Sky Imager. See **Line 41 - 51** in the revised manuscript.

- **Lines 46 - 48**: "Details on the observation mode (including the temporal resolution and integration time) and production of the final image can be seen elsewhere in Bageston et al. (2009) and Nyassor et al. (2021, 2022)."

  **Question:** Temporal and horizontal spatial resolutions are not "details" but the main parameters of the instrument, which should be presented here.

  **Response:** Based on your comment, the entire section of the all-sky imager and the observation mode has been written all over in the main text as shown in the red highlighted text below:

This imager is equipped with a Charge Coupled Device (CCD) camera (SBIG, STL-1001E model), which has a resolution of $1024 \times 1024$ pixels, each pixel measuring $24.6 \times 24.6 \,\mu$m, and 50% of quantum efficiency in the near-infrared spectrum. The image was not binned but cropped to $512 \times 512$ pixels, producing a final image size of $12 \times 12$ mm on the CCD chip with a spatial resolution of $512 \times 512$ km. Each image has an integration time of 20 s and a readout time of 10 s, since the imager does not have a filter wheel, the temporal resolution is 38 seconds (Bageston et al., 2009; Nyassor et al., (2021,2022)). Airglow observations were taken when the Sun and Moon elevations were lower than -12° and 10°, respectively. This mode allows 28 nights of observation per month, centered on the new moon.

This text can be seen on **Line 41 - 51** in the revised manuscript.

- **Line 66-67**: "Afterward, the images are unwarped and mapped into the geographical coordinates." It is unclear how were images mapped into the geographic coordinates? In order to do it, one needs to make a total geometrical calibration (not only the alignment of the original airglow image to the geographical north) but solving the optical model of the camera for EACH pixel. A brief procedure of the geometrical calibration for each pixel is needed here.

- **Lines 76-77:** "Figure 1. Flowchart showing the procedures of airglow image processing and wave parameter estimation. The three stages describe image preprocessing and processing, spectral analysis, and wave parameters estimation procedure." In the flowchart and in section 3 "Methodology and Data Analysis", I cannot see removals of the atmospheric background (not atmospheric extinction) and noise level of the sensor. Also, there is no information on the flat field correction of the sensor. How these issues have been treated in the image processing? These should be described.

  **Response:** Due to your comment, the entire section of the methodology has been written all over the main text, as shown in the red highlighted text below:

[revised manuscript text omitted]

This text can be seen on **Line 61 - 97** in the revised manuscript as well as the revised Flowchart.

[Figure]

Figure 1: Flowchart showing the procedures of airglow image processing and wave parameter estimation. The three stages describe image preprocessing and processing, spectral analysis, and wave parameters estimation procedure.

- Before presenting statistical results, it is worth presenting a figure demonstrating an example of an observed quasi-monochromatic gravity wave.

  **Response:** The figure of the observed QMGWs and keogram presented in the main text is shown below.

[Figure]

Figure 2: The observed quasi-monochromatic gravity wave (QMGW) on July 20 to 21 2017 at São Martinho da Serra.

Below is the description of the 2 as it is in the revised manuscript:

In Figure 2, a sample of four preprocess images at 23:00:01 UT, 00:00:49 UT, 01:00:16 UT and 02:01:04 UT plotted on the geographic map is presented in panels (a), (b), (c), and (d), respectively. The white triangle with a black outline shows the center of the all-sky image and the location of the imager. The gray lines in panels (a), (b), (c), and (d) depict the state borders of Rio Grande do Sul. The bright strand extending from the southwest through the center of the image to the northeast is the

Milky Way. In the keogram, the Milky Way is the white strand extending through the middle throughout the observation. It is important to mention that the keograms presented here are only used to show the presence of the QMGWs throughout the observation time from 21:00 UT on July 20$^{\text{th}}$ to 09:00 UT on 21$^{\text{st}}$. It can be seen that the wave packet changes with time from the northwest to the southwest. An animation of the propagation of the 20 - 21, 2017 QMGW event between 21:00 UT on July 20 2017 and 09:00 UT on July 21, 2017 is provided in the video supplement. In panels (e) and (f), the zonal and meridional components of the keogram, where a downward phase progression of black and white undulations can be seen in the zonal component. The black and white vertical undulations have an upward phase progression in the meridional component of the keogram. This clearly shows the presence of a quasi-monochromatic structure throughout the night.

- Something is wrong with the caption on the horizontal axis of Fig. 3c.

  **Response:** The caption of Figure 3c (now 4c) has been corrected and the caption of entire figure has been revised as shown below:

  Quasi-Monochromatic Gravity Waves (QMGWs) characteristics over five years of observations at São Martinho da Serra. Panels (a), (b), and (c) present the histogram of the horizontal wavelength ($\lambda_H$), period ($\tau_H$), and phase speed ($c_H$), respectively. In panel (d), the propagation direction ($\phi$) of the QMGWs are presented according to season.

  See Figure 4 and Caption on page 9 of the revised manuscript.

- **Line 153 - 154:** "Similarly, most waves observed in the OH airglow images were visible and propagated for 2 - 3 hours."

  **Question:** I do not understand this definition "duration of propagation of the waves in the OH images´´. Does it mean that an observed wave package (a number of wave crests and troughs) did propagate for 2 - 3 hours in the field of view of the imager? Or is it something else? This should be explained.

  **Response:** We meant to say observed wave packet (a number of wave crests and troughs) did propagate for 2 - 3 hours across the field of view of the imager as you rightly said. So, according to your comment, the text below has been added for clarification.The new added texts are in blue.

  The propagation time of the waves from their source to the observation altitude in the mesosphere is presented in Figure 6(a), while the duration of propagation (thus the time span the propagating waves were visible in the OH images during the night) of the waves in the OH images is presented in Figure 6(b). It can be seen that the majority of the wave propagated less than one hour from the source position in the lower atmosphere to the OH emission layer. Similarly, the observed wave packet from which the individual QMGWs were selected in the OH airglow images were visible over the field of view of the all-sky imager and propagated for 2 - 3 hours.

  See Line 196 - 197 and Line 198 - 200 of the revised manuscript for this chanages.

- **Lines 406 - 407:** "Also, Mastrantonio et al. (1976) showed that the gravity waves generated by tropospheric jet streams may have the ability to propagate vertically to the upper atmosphere, such as the ionosphere (Mastrantonio et al., 1976).´´

**Here it is worth citing two more papers:**

Dalin et al. (2015) demonstrated a particular transient isolated gravity wave in the summer mesopause associated with the passage of an occluded front and/or the point of occlusion. The mechanism of the wave generation was likely due to strong horizontal wind shears at about 5 km altitude. Dalin et al. (2016) demonstrated that gravity waves, observed in the summer mesopause, were associated with the upper tropospheric jet stream at altitudes 8 – 10 km.

**Additional references:**

Dalin, P., A. Pogoreltsev, N. Pertsev, V. Perminov, N. Shevchuk, A. Dubietis, M. Zalcik, S. Kulikov, A. Zadorozhny, D. Kudabayeva, A. Solodovnik, G. Salakhutdinov, I. Grigoryeva: Evidence of the formation of noctilucent clouds due to propagation of an isolated gravity wave caused by a tropospheric occluded front. Geophysical Research Letters, 42, 2037-2046, doi:10.1002/2014GL062776, 2015.

Dalin, P., N. Gavrilov, N. Pertsev, V. Perminov, A. Pogoreltsev, N. Shevchuk, A. Dubietis, P. Völger, M. Zalcik, A. Ling, S. Kulikov, A. Zadorozhny, G. Salakhutdinov, and I. Grigoryeva: A case study of long gravity wave crests in noctilucent clouds and their origin in the upper tropospheric jet stream. Journal of Geophysical Research - Atmospheres, 121, doi:10.1002/2016JD025422, 2016.

**Response:** Thanks very much. The references has been in cited in the revised manuscript as shown below:

Using two synchronized automated digital cameras at Krasnogorsk and Obninsk, located near Moscow, Russia, Dalin et al. (2015) demonstrated that a particular transient isolated gravity wave in the summer mesopause is associated with the passage of an occluded front and/or the point of occlusion. The source mechanism of the wave generation, according to Dalin et al. (2015), was likely due to strong horizontal wind shears at about 5 km altitude. Similarly, Dalin et al. (2016) illustrated that gravity waves, observed in the summer mesopause, were associated with the upper tropospheric jet stream at altitudes 8 – 10 km.

---

## Author Comment (AC2)

**Referee #1 Evaluations:**

**Reviewer #1 (Formal Review for Author)**

This paper provides statistics on quasi-monochromatic gravity wave events from 2017 to 2022, based on the observation of all-sky airglow imager. The propagation parameters of the waves were extracted using 2D Fourier transform, and the positions of wave sources were inferred with the ray tracing model. In three case studies, the wave sources were related to jet streams, instead of deep convection. This paper is based on abundant observation data and logically-organized, so I think this should be acceptable. However, I have two small questions as below:

**Appreciations:** The authors would like to thank the reference for taking time off to review this manuscript and for the comments, suggestions, and corrections. We have given the response accordingly.

- In the abstract (**line 1**) and conclusion (**line 463**), the authors said there were 209 events found during the five years. But in **line 114-115**, the number of gravity wave cases changed to 64. Why are there two different numbers?

  **Response:** Here, we intend to say that 64 nights out of the 1512 clear sky nights were the nights from which the 209 individual Quasi-Monochromatic Gravity Waves events were obtained. However, as you have pointed it out, we have rewritten this part as:

  A total of 1512 nights of clear sky images were analysed. From 64 nights, 209 QMGWs events were obtained [Line 156 - 157].

- How did the authors find out the total 209 (or 64) gravity wave events from the airglow images captured in the five years? Have any methods been taken to avoid omissions and misjudgments?

  **Response:**

  Before selecting the QMGWs, the waves must first and foremost be visible in the OH image for not less than two(2) hours. Next, the wave parameters were determined every ten (10) minutes. This is done to track the variations in the wave parameters (specifically the horizontal wavelength) to make sure that it is the same wave. GWs that satisfied this criteria are chosen to go through the next stage of criteria. The above described point are included in the main text as:

  The five years of observed OH airglow images were subjected to spectral analysis to estimate the QMGW characteristics. Specific criteria were imposed to select the QMGW events used in this work. After the spectral analysis, the confidence level (CL) of the estimated wave spectrum is estimated. The spectrum having peak power spectral density with CL $\geq 95\%$ is accepted (Hu et al., 2002). Before selecting the QMGWs, the waves must first and foremost be visible in the OH image for not less than two (2) hours. Next, the wave parameters were determined every ten (10) minutes. This is done to track the variations in the wave parameters (specifically the

horizontal wavelength) to make sure that it is the same wave. If the conditions; CL $\geq$ 95%, visibility of wave and the determined wave being similar are satisfied, the wave is then subjected to the following conditions: [Line 166 - 170]

i. the $\lambda_H$ must be greater than or equal to 10 km ($\lambda_H \geq 10$ km);

ii. the variation of $\phi$ within an hour must be less than 25° ($\Delta\phi \leq \pm 25°$);

iii. the GW propagation period must be between 5 and 80 min (5 min $\geq \tau_H \leq$ 80 min), and

iv. finally, the GW phase speed must be less than or equal to 150 m/s ($c_H \leq 150$ m/s).

5

**5Response:** Based on your comment, the entire section of the all-sky imager and the observation mode has been written all over in the main text as shown in the red highlighted text below: